# Trends and Developments in Medical Liability Claims in The Netherlands [note 1]

**DOI:** 10.3390/healthcare10101929

**Published:** 2022-10-01

**Authors:** Désirée Klemann, Helen Mertens, Frits van Merode

**Affiliations:** 1Department of Gynecology and Obstetrics, Maastricht University Medical Centre+, Maastricht University, 6229 HX Maastricht, The Netherlands; 2Care and Public Health Research Institute, Maastricht University, 6200 MD Maastricht, The Netherlands; 3Executive Board, Maastricht University Medical Centre+, Maastricht University, 6229 HX Maastricht, The Netherlands; 4Maastricht University Medical Centre+, Maastricht University, 6229 HX Maastricht, The Netherlands

**Keywords:** malpractice claims, medical liability, litigation, trends, developments, COVID-19, seasonal fluctuation

## Abstract

Recent data on number of claims, final judgement of claims and their costs are scarce. This study analyzes 15 years of malpractice claims in the Netherlands. All claims filed, and all claims closed by two insurance companies (which insure approximately 95% of all hospitals in the Netherlands) between 2007–2021 are included. Trends in number of claims, medical specialties involved, final judgements and costs from malpractice claims are analyzed, as well as the impact of COVID-19 on malpractice claims. In total, 20,726 claims were filed and 21,826 claims were closed. Since 2013, the number of claims filed decreased. Of all claims filed, 64% were aimed at surgical specialties and 18% at contemplative specialties. Of all claims closed, 24.49% were accepted, 19.26% were settled and 48.94% got rejected. The financial burden of all claims closed quadrupled between 2007 and 2021; this increase was caused by rare cases with excessively high costs. Since the COVID-19 pandemic, we observed a decrease in the number of claims filed, and the number of incidents reported. This study provides valuable insights into trends and developments in the number and costs of liability claims, which is the first step towards improving patient safety and preventing incidents and malpractice claims.

## 1. Introduction

Over the last decades, medical malpractice claims have acquired increased public attention, especially in the U.S.A. In the early 2000’s in the U.S.A. a true ‘malpractice crisis’ took place, meaning that the number and costs of malpractice claims increased rapidly, leading to untenable high insurance premiums [1]. Available claim data from the U.S.A. are derived from the National Practitioner Data Bank (NPDB) and describe the number and amount of claims paid on behalf of physicians [2]. Unfortunately, they do not provide any insight into the number of claims submitted but rejected and/or trends in the amount of claims submitted. 

European physicians, insurance companies and policy makers fear a similar increase in number and costs related to medical malpractice claims in Europe. A recent published overview of national trends and costs of litigation in the United Kingdom’s National Health Service seems to confirm this [3]. Annual claim costs in the United Kingdom have quadrupled since 2006 and ran up to GBP 3.6 billion in 2018/2019. Based on these data, beside specific risk factors for injury and malpractice claims per medical specialty, certain general factors are described that might contribute to the risk of malpractice claims. For example, a catastrophic healthcare outcome after treatment, ineffective doctor–patient communication (and a lack of informed consent) and setting unrealistic treatment targets and expectations [3].

Although these factors may contribute to the occurrence of incidents and malpractice claims, they do not explain the increase in number and costs of malpractice claims. Did the healthcare system, despite numerous initiatives to improve patient safety and quality of healthcare, get more dangerous, leading toward more medical incidents? Or are we (possibly due to a change in economic and financial situation, or as a result of increased notoriety of claims) heading towards a ‘claim culture’, in which people in general, and patients specifically, tend to submit claims more easily? Or has a change in claim assessment and judgement resulted in higher damage compensation and costs?

In order to answer these questions, this study analyzes 15 years of malpractice claims on a national level in the Netherlands. This research will gain insight into trends and developments in the number of claims submitted, closed and payments from malpractice claims from 2007 to 2021. In contrast to other studies, we will not only focus on claims in which the patient received a financial compensation, but we will include all claims submitted, even the rejected claims. In this way, we gain further insight into the reported incidents, claims submitted and their final judgement (accepted, rejected, settlement made). Second, we will analyze the distribution of costs from malpractice claims. Is there a (relative) increase in overhead costs? Are costs and damages paid increasing overall, or in specific categories of claims?

During the last two included research years, the size, type and severity of delivered hospital care changed radically as a result of the COVID-19 pandemic. Some fear that the pandemic will cause an increase in complaints and claims, since volumes of patients increased rapidly, a delay occurred for most (elective) care and hospitals, doctors and nurses faced logistical and medical challenges [4,5]. We will analyze the impact of COVID-19 on the number and subject of medical liability claims.

## 2. Materials and Methods

### 2.1. Study Design

This study is a quantitative, descriptive and evaluative study at a national level. In the Netherlands, hospitals and their employees, including doctors, nurses and other (paramedical) staff, are collectively insured for liability claims. Doctors and other hospital staff usually do not have a separate insurance for malpractice claims. In 2021, the Netherlands counted 82 hospitals (categorical, general and university centers). Of these hospitals, 76 (93%) are insured for medical malpractice claims by insurance companies Centramed or MediRisk.

To evaluate medical liability claims in the Netherlands, we used the databases from Centramed and MediRisk. Anonymized claim data were obtained from both insurance companies in a uniform manner.

### 2.2. The Medical Liability System in The Netherlands

If an (alleged) medical error occurs or when people are dissatisfied with the quality or outcome of their medical treatment, patients have several options to express this dissatisfaction. All options serve different purposes, for example recovery of the relationship between doctor and patient, improving quality of healthcare, sanctioning of the doctor or providing a financial compensation for medical damage.

The first, most accessible option is provided by a ‘complaints officer’ (1), which every hospital is obliged to have since 2016. The complaints officer is an independent employee and aims to mediate between doctor and patient, in order to regain trust between doctor and patient. If this mediation process is successful, the complaints officer might inform and advise the patient through other ways to express dissatisfaction. Since 2016, with the introduction of a new law (Wkkgz), every hospital has been affiliated with an independent Complaints Committee (2), which will try (again) to mediate between doctor and patient, but can also award a financial compensation up to EUR 25,000. Before a patient can file a complaint at this Complaints Committee, a mediation process by the complaints officer is mandatory. In practice, there are very few complaints submitted by this Complaints Committee (in 2021 the Complaints Committee awarded a financial compensation in only six cases; costs per case amounted between EUR 150 and EUR 7500).

In addition, a patient has the possibility to submit a complaint to a medical disciplinary committee (3). The primary aim of this disciplinary committee is to guarantee the quality of the medical profession. It can for example impose penalties or (temporarily) remove a doctor from his/her profession. The disciplinary committee is not able to award financial compensation for material and immaterial damage a patient might have suffered.

If a patient aims a financial compensation for (im)material harm due to a medical mistake, the patient should file a liability claim (4) against the doctor or the hospital they were treated in. Medical liability claims are mostly based on article 6:74 of the Dutch Civil Code (DCC), which holds a ‘failure to correctly comply a contractual obligation’. If a patient submits a liability claim against a hospital or its employees, the hospital will most often forward the liability claim to the liability insurance company (such as Centramed and MediRisk). With the help of expert advice, the insurance companies will make a judgement about liability and causality between the alleged error and damage suffered. If liability is acknowledged, the patient will receive a financial compensation for the damage suffered. When the parties fail to agree on the question of liability or the amount of compensation to be paid, the patient can submit the case to a civil court. Approximately 95% of all liability claims in the Netherlands are handled out-of-court. 

Finally, in rare cases if the patient believes that the doctor committed a criminal offence, he/she can report the doctor after which criminal proceedings [5] may follow. These rare cases, which occur on average once a year [6], fall outside the scope of this study.

### 2.3. In- and Exclusion Criteria

This study focusses on medical liability claims submitted by the patients to hospitals and handed over by the hospitals to their medical liability insurance companies (Centramed or MediRisk). In order to gain insight into the number of claims made, as well as the closing process, judgments and costs of claims closed, we selected two separate study groups. Group A includes all hospital-related claims filed by Centramed and MediRisk between 1 January 2007 and 31 December 2021. Group B includes all hospital-related claims closed by Centramed and MediRisk between 1 January 2007 and 31 December 2021, regardless of the date the claims were filed. All data were collected using reference date 31 December 2021. 

In order to discern developments and trends in the number and extent of claims, we had to maintain the group of affiliated hospitals constantly through the research period. Hospitals that merged, or hospitals that switched between the two insurance companies within the research period, were included. All claims related to hospitals that were not insured by Centramed and/or MediRisk for the entire duration of this study were excluded (N = 5). In total, we analyzed all claims submitted and closed for 71 (87%) of all hospitals in the Netherlands. We excluded claims regarding other healthcare institutions (such as pharmacies or psychiatric institutions).

### 2.4. Extent of Care Provided per Medical Specialty

In order to relate the number of medical liability claims and costs with the size of this medical specialty, we calculated the financial turnover per medical specialty per year. To calculate the financial turnover, we used a freely accessible database from the Dutch healthcare authorities [7]. This database includes the number of patients, diagnoses, treatments and costs per treatment per patient per specialty. We used these numbers to calculate the financial turnover per medical specialty for the year 2012 (the year in the middle of the research period, the COVID-19 years excluded). The financial turnover per specialty helps us to estimate the scope of each medical specialty, but does not provide insight into the complexity of delivered healthcare per medical specialty. 

### 2.5. Definitions

We analyzed the number of claims filed and closed per medical specialty. We categorized these medical specialties into the following categories: surgical, contemplative, supporting, paramedical, unknown and others (see Table 1). 

The reporting time is the time (days) between medical incident and submission of claim.

The insurance companies make a final judgement in each liability claim. Final judgements are classified as ‘accepted’, ‘rejected’ or ‘settlement made’. An accepted claim means that liability is acknowledged and generally the patient receives a financial compensation for the damage suffered as a result of the medical mistake. A rejected claim means that liability is disclaimed because there is no medical mistake, or a lack of causality. In case a settlement is made, the insurance company does not acknowledge liability, but nevertheless offers the patient a financial compensation for medical damage. In practice, this is a common method to avoid long-term legal procedures. 

The costs associated with the claims closed are described below as ‘compensation paid to patients’ (sometimes including a ‘lump sum’) and ‘total costs’. The compensation paid to a patient is the total amount of money the patient received to compensate for the (medical) damage suffered. The total costs include, beside the compensation paid to the patient, also the costs associated with the settlement of the liability claims, such as legal costs and costs for external expert opinions. Operating costs and file costs of the insurance companies fall beyond the scope of this study.

### 2.6. Analyses

All data from Centramed and MediRisk have been supplied in Microsoft Excel [8]. The software used for the analysis is: Python 3.8.6 [9], Pandas [10,11], Matplotlib [12,13], Statsmodels [14] and NumPy [15]. 

## 3. Results

### 3.1. Trends and Developments in Claims Filed (Group A; Reporting Year)

Between January 2007 and December 2021, 20,726 claims were filed, with an average of 1382 per year (varying from 972–1711). Figure 1 demonstrates trends of claims filed per year with an increasing trend from 2007 until 2013, after which a decrease is observed, with the lowest number of claims filed in 2021. 

#### 3.1.1. Claims per Medical Specialty

Of all claims filed, 64% (53–68) are aimed at surgical specialties, followed by 18% contemplative specialties (14–25) and 11% supporting specialties (9–14). This ratio does fluctuate a little bit, but does not significantly change during the years.

#### 3.1.2. Reporting Time

We analyzed the reporting time per medical specialty and per category specialties. Overall, the mean reporting time is 757.84 days. The median time is 436 days. Figure 2 shows that the mean and median reporting time is almost similar for surgical and contemplative specialties. This means that most submitted claims relate to incidents that occurred 1.5–2 years before.

In the boxplots below (Figure 3), the total number of incidents reported (2007–2021) per year and month in which the incident occurred is shown (note: this concerns the incident year, not the month/year the claim was submitted).

As can be observed there is a downward trend of incidents during the period 2007–2021. This downward trend may (partially) reflect the time lag between incident and submission of claim. In addition, the distribution of incidents during the year suggests a seasonable pattern, to which we come back below. During the last year there is underreporting of incidents because of the time lag between the incident and reporting it. 

The number of incidents reported in 2020 are considerably lower than the years before (lower than the first quartile), and lower than expected based on the downward trend. This might be an effect of the COVID-19 pandemic that had a massive impact on Dutch society and on type, size, complexity and amount of hospital care. To get more insight into trends and thereby also in the ‘COVID-19’ effects, we selected all cases (N = 4721) with claims submitted within 183 days (6 months) after the date of incident before 1 July 2021. In this way we could compare the trend of incidents during the years, filtering out the effect of delay between incident and claim within the period of 15 years. The drawback is that we are not sure about the trend of incidents reported longer than 183 days after the incident. Figure 4 shows the number of reported incidents within 183 days after the date of incident. As can be observed there is—again—a downward trend, with substantial fluctuations.

Figure 5 shows a year- and month wise box plot for all claims submitted within 183 days after the date of the incident from 2007–2021. Once again, an impressive decrease of reported incidents in 2020 can be observed. 

### 3.2. Trends and Developments in Claims Closed (Group B; Closing Year)

Between January 2007 and December 2021, 21,826 claims were closed, with an average of 1455 per year (varying from 1120–1894). Of these claims closed, 65.08% were aimed at surgical specialties, 17.13% at contemplative specialties and 11.62% at supporting specialties. During the years, this ratio fluctuates but stays in margins. 

#### 3.2.1. Final Judgment of the Claims Closed

A total of 24.49% of all claims closed got accepted and the patient received an amount to compensate the damage suffered. In 19.26%, a settlement was made. In almost half of all claims closed (48.94%), the claim was rejected. This ratio did not change significantly during the research period. Figure 6 shows the ratio of the final judgement per category of medical specialty.

#### 3.2.2. Financial Burden of the Claims Closed

The financial burden of these claims closed amounts to EUR 404,155,330, of which 74% (EUR 297,760,00) was paid to the patient. Figure 7 shows the total costs and compensation paid to the patient per year (2007–2021). The mean burden per claim closed amounts to EUR 18,520. For claims which were accepted the mean costs were EUR 56,838 and the median costs were EUR 20,975. The mean total payment to patients was EUR 43,942 and the median was EUR 13,750. For settled claims the mean total cost was EUR 19,707, and the median was EUR 6403. The mean payment to patients for settled cases was EUR 14,520 and the median was EUR 4000. The claim with the highest damage burden was closed in 2017; the total costs of this claim amounted to EUR 3,106,835.

In most claims (41%), costs amount to less than EUR 1000. In 33% of claims, costs amount to EUR 1000–EUR 10,000, in 10% EUR 10,000–EUR 25,000, in 10% EUR 25,000–EUR 100,00, and in only 4% of claims, costs amount to more than EUR 100,000. 

Figure 8 shows the mean, median, first and third quartile of the total costs per claim per year. The median costs per claim remain quite stable, while the mean costs are rising, indicating that there are few (accepted) claims with excessively high costs. The difference between mean and median costs is increasing each year with the exception of the years 2019–2021. It is remarkable that the mean costs in 2019 are relatively low.

#### 3.2.3. Claims per Medical Specialty

The top three medical specialties with the highest number of claims filed and the highest damage burden are similar each year, knowing: (1) general surgeon, (2) orthopedic surgeon and (3) gynecology/obstetrics. Table 2 shows the top 10 specialties with the highest damage burden, related to their share in the annual financial turnover (based on year 2012).

## 4. Discussion

We use malpractice data to observe trends and developments in number of claims submitted, closed, medical specialties involved and costs from malpractice claims. The financial turnover per medical specialty was used to relate malpractice data with the size of each medical specialty. We also use the malpractice claims as a derivative of reported incidents. Of course, not every incident leads to a malpractice claim, and not in every claim submitted, a real medical incident occurred. Nevertheless, malpractice claims provide a representative insight into the quality of care as experienced by patients.

### 4.1. Number of Claims Filed and Number of Incidents

After the before mentioned ‘malpractice crisis’ [1], U.S.A. malpractice data showed a 55.7% decreased ratio of paid claims, from 20.1 per 1000 physician years in 1992–1996, to 8.9 per 1000 physician years in 2009–2014 [2]. These data do not provide any information about the number of claims submitted but rejected and without financial compensation. Data from the U.K. do not provide national trends in the number of claims submitted, but do describe an increase in the number of claims submitted. The increase in the number of claims submitted varies per medical specialty. Surgical, medical, acute and obstetric claims increased most in 2019 compared to 2009. Understanding the differences between countries and comparing trends and developments is important to gain insight and to learn. With this study, we want to contribute to comparative studies.

This study does not confirm an increase in the number of claims submitted in 2021, compared to 2007 [16]. In the first 5 years of this study, from 2007 until 2013, an increase of claims submitted can be observed. From 2013 and later, the number of claims submitted decreased. All medical specialties followed the same trend. The number of incidents reported per year also decreased during the last decade. Although this downward trend may (partially) reflect the time lag between incident and submission of claim, we also observed a decreased number of claims reported within six months of the date of incident. Increased attention to quality improvements may have reduced the risk of adverse events and thus led to a decreasing trend of malpractice claims. The downward trend in number of claims filed argues against an upcoming ‘claim culture’. 

However, a relocation of claims cannot be excluded, since liability insurance companies encouraged affiliated hospitals to settle relatively small and straight-forward claims by themselves. To stimulate this, a ‘deductible’ was introduced, meaning that the insured hospitals have to pay a certain amount when they file a malpractice claim at the insurance company. Second, the introduction of the before mentioned law in 2016 (Wkkgz) stimulated complaint and claim settlement close to the source, by mandating mediation by a complaints officer. This may have led to an improvement in complaint handling and thus a decrease in malpractice claims. Third, since the introduction of the Wkkgz, patients have a choice of submitting their claim to the hospital, or to an independent complaints committee, which can award reimbursements of up to EUR 25,000. However, if these developments would have influenced the number of claims submitted to the insurance companies, this should have resulted in a decrease of number of claims with a low claim burden. This cannot be observed in the analyses of the claims closed, their final judgement and the costs related to the claims, since the ratio within the final judgments remained unchanged and the costs have only increased over the last 15 years. 

### 4.2. Medical Specialties Involved

Medical liability claims are registered and settled per individual healthcare worker, both in and out-of-court. Similar to the findings in the U.S.A. and the U.K., most (two-thirds) of all claims submitted and closed are related with surgical specialties, versus 17% contemplative specialties and 12% supporting specialties. This does not automatically mean that two-thirds of all claims are related to perioperative incidents. Surgical specialties can, of course, cause diagnostic or medication related incidents as well. Nevertheless, a Japanese study [17] of machine learning-based prediction models for litigation outcomes found that 41.67% of all reported incidents occurred in an operation room. Procedures or surgeries were the most common reasons for litigation, with also the highest acceptance rate (56.1%). Possibly, whenever a perioperative incident occurs, the relation between surgery, outcome and involved doctor is more obvious and therefore more prone for malpractice claims, compared to diagnostic incidents. With diagnostic errors, the error is often recognized after a longer time lag, when the hypothetical health outcome without the error is more difficult to reconstruct, and the responsible doctor more difficult to identify, therefore leading to less malpractice claims. 

Each year, the same medical specialties form the top three in terms of number of claims submitted and highest costs: general surgery, orthopedics and gynecology/obstetrics. We calculated the financial turnover for each medical specialty and used this as a derivative for the production per specialty. As presented in Table 2, although general surgery, orthopedics and gynecology have a large share in the financial turnover, this is not proportional to their share in costs of malpractice claims. Possible explanations for the relatively high share in number of claims and costs from these specialties might be the surgical character, a large part of elective care (in which a patient might be more disappointed when healthcare outcomes are not as expected, compared to oncological and emergency care). This should be a topic for further research.

### 4.3. Final Judgement and Costs of Claims

None of the previous studies provide insight into the final judgement of claims, since they only describe the claims in which a financial compensation was paid. Our study shows that the majority—almost half—of malpractice claims got rejected. A quarter of all claims closed got accepted and in 19% a settlement was made. The relative number of settlements increased slightly between 2007 (14.24%) and 2021 (20.79%). Again, these findings argue against an upcoming ‘claim culture’, because in a ‘claim culture’ more ‘trivial claims’ would be expected, leading to a larger share of rejected claims [18].

It is remarkable that, despite the decreased number of claims filed and a stable number of claims closed, the costs of malpractice claims increased rapidly, showing the same increasing trend as in the U.K and U.S.A. [2,3]. Total costs of claims quadrupled in the Netherlands between 2007 (EUR 9,029,850) and 2021 (EUR 40,938,960). In the U.K., the total annual cost of litigation amounts to GBP 3.6 billion, of which GBP 1.5 billion was to compensate damages. In the Netherlands, 74% of all costs are paid to the patients to compensate damage. In the Netherlands, for claims which were accepted the mean costs were EUR 56,838 and the median costs were EUR 20,975. These amounts are significantly lower compared to mean and median costs in the U.K. and U.S.A. The mean compensation payment in the USA in 2009–2014 amounted $ 353,473. We would suggest further research into the nature of these compensation payments and possible differences between the Netherlands and for example the U.S.A. and the U.K. In the Netherlands, medical incidents with a permanent invalidity will often lead to a higher damage burden compared to an avoidable death, since in case of invalidity, future healthcare costs (e.g., nursing costs, medical supplies) and loss of income will also be covered by the insurance companies. Compensation for affection damage is still relatively low in the Dutch liability system and jurisdiction.

As can be observed in Figure 8, median costs per claim did not significantly increase between 2007 and 2021. There is an increase of rare cases with excessively high costs that increase the total and mean costs for malpractice claims. These rare cases are claims related to incidents with serious and permanent healthcare damage, for example, birth related damages. An increase of these rare cases and their costs may be due to jurisdiction and/or the attendance of injury lawyers. Social-economical changes, which lead to decreased social assistance or support, may also increase the costs that have to be compensated by the insurance companies. 

### 4.4. The Impact of COVID-19

Further increase of malpractice claims and their costs as an effect of the COVID-19 pandemic, for example as a result of higher volumes of patients attending Accident and Emergency Departments and a delay in elective healthcare [4,5] did not materialize. Since March 2020, the COVID-19 pandemic had a massive impact on Dutch society. There is no information available yet describing the effect of COVID-19 on malpractice claims. Although it is still early to set statements on this subject (especially considering the time gap between incidents and moment of reporting them), we found a noticeable decrease in number of claims filed in 2020 and 2021. This decrease was bigger than expected based on the previously declining trend. 

The pandemic necessitated social measures, such as restrictions on freedom and a lockdown. These measures could have caused a delay in submitting claims, making it more difficult for people to contact complaints officers, medical experts or legal aids. Since the pandemic, the societies’ perspective on healthcare workers seemed (temporarily) changed: people had, especially in the beginning of the pandemic, more compassion and admiration for healthcare workers, which may have influenced the willingness to submit a claim against healthcare workers. Besides these social changes, COVID-19 changed the type of hospital care radically; all elective care, scheduled surgery and treatments were temporarily postponed in order to retain capacity for COVID-19 suffering patients. This change of type and size of care may have led to a changed risk of incidents, and therefore a decrease of claims. 

The median reporting time between incident and submission of claims amounts up to 1.5–2 years. This implies that a changed risk of incidents as a result of the changed size and type of care during COVID-19 pandemic cannot be fully measured in the first years after the pandemic. Still, there are claims with a relatively small reporting time. To analyze possible trends in reported incidents as an effect of the COVID-19 pandemic without the effect of delay between incident and submission of claim, we analyzed the number of incidents reported within 183 days after the date of incident between 2007 and July 2021. The results are shown in Figure 4 and Figure 5. The number of incidents show, similar to the number of claims per year, a decreasing trend. In 2020, the number of incidents is even lower than expected based on the decreasing trend. This means that the risk of incidents (or the willingness to submit a claim!) decreased as a result of the changes in hospital care during the pandemic. In the first six months of 2021, the number of reported incidents was back to the level expected based on the trendline. This argues against deferred claim filing as a result of social measures. 

### 4.5. Seasonal Fluctuation

Analyzing the trends in number of claims submitted and in reported incidents per year and per month, we did not only observe a downward trend; we also found a remarkable fluctuation in reported incidents per month. Therefore, we further decomposed these findings to discern trends and seasonal fluctuations, leading to Figure 3 and Figure 5. We discovered an annual seasonal fluctuation, both in all reported incidents as well as in the incidents reported within 183 days after the date of the incident. Each year, most incidents occur in December and January, while in summer a notable decrease of incidents occur. During the summer months, less variation in incidents is also observed. This phenomenon has not been described before and literature does not provide any explanation for this seasonal fluctuation. Possible explanations may be seasonal health conditions, seasonal variation in health supply, such as a reduction of treatments and surgery during summer months or staff-related fluctuations. Given the striking recurrent pattern, further research into this phenomenon is indicated. 

### 4.6. Strengths and Limitations

This study provides valuable insights into trends and developments in the number and costs of liability claims, which is the first step towards improving patient safety and preventing incidents and malpractice claims. Considering the high coverage ratio of the insurance companies Centramed and MediRisk, and the broad scope of this study (including all types of hospitals and all medical specialties), our data provide a reliable overview of national trends and developments in malpractice claims in hospitals in the Netherlands over a 15-year period. Since we excluded hospitals that were not insured by Centramed or MediRisk for the entire period of the study, trends and developments found cannot be attributed to a changed market share of the insurance companies. It is also unique that all claims submitted are analyzed, not only the cases in which liability was accepted or a settlement was made. 

Unfortunately, our data only provide insight into trends and developments in malpractice claims, submitted to the insurance companies and related to hospital care. Claims handled by the hospitals themselves (as an effect of deductible costs) are beyond the scope of this study, as are all malpractice claims in non-hospital care, such as general practices or nursing home care. In addition, it is uncertain how many of the included claims were (also) taken to court, and if these claims differ in character, outcomes and costs compared to the claims only handled by the insurance companies. Whether a case is taken to court, depends not only on the (potential) size of a claim, but also on the ruling settlement culture, and that may vary per country. At last, in order to gain insight into the subject of claims, type of incidents (for example diagnosis errors vs. surgical errors) and claims related to team professional liability (incidents related to collaboration between colleagues and different or multiple medical specialties), a further analysis of claim files would be necessary.

## 5. Conclusions

An increase of malpractice claims and a commencing claim culture are often feared. This study on national level shows the opposite: there is a decreasing trend in number of claims submitted, arguing against a claim culture. In addition, based on malpractice claims, there is a decreased number of incidents reported over the last 15 years, which may point to the positive effect of quality improvements in hospital care. At the same time, malpractice claim costs are rising rapidly, caused by few claims with excessively high costs, while in most claims, the median costs and damages paid remain quite stable. Every year the same medical specialties (general surgery, orthopedics and gynecology) form a top three regarding the number of claims and costs. 

This is the first study to describe the effect of COVID-19 on malpractice claims. We observed a remarkable decrease of claims submitted and incidents reported since COVID-19 influenced hospital care and caused social measures. This effect lasts even when we correct for delay between incident and submission of claim, by focusing on claims reported within 183 days after the date of the incident. If this would be due to deferred claim submission because of social measures, we would expect an increase of claim submission in 2022, which cannot be observed. It is possible that the decrease of claims and incidents is related to the change in type and size of hospital care (less elective care, more COVID-19 related care). Another explanation for the decrease of claims and incidents reported may be a positively changed social perspective on healthcare workers, whereby the preparedness to submit a claim is reduced. Because of the known delay between incident and submitting a claim and to exclude the consequences of deferred claim submission, we advise to analyze the number of claims submitted and incidents reported during the COVID-19 pandemic again after two years. 

A noteworthy finding during our analysis is a yearly repeated, steady seasonal fluctuation in incidents reported. Of course, not every incident leads to a liability claim, and a medical incident did not occur in every submitted claim, but still this fluctuation should be further analyzed, since it can point into valuable information regarding causes of incidents or medical liability claims. Possible causes for seasonal fluctuation could be, for example, related to production conditions and supply of healthcare, personal staff or seasonal healthcare conditions. Further depth analysis of this seasonal fluctuation of incidents and depth analysis of claim files is recommended to reveal possible clinical patterns that contribute to medical errors and enable physicians, policy makers and insurance companies to improve healthcare, invest in patient safety and to invest trends of financial consequences of malpractice claims.

## Figures and Tables

**Figure 1 healthcare-10-01929-f001:**
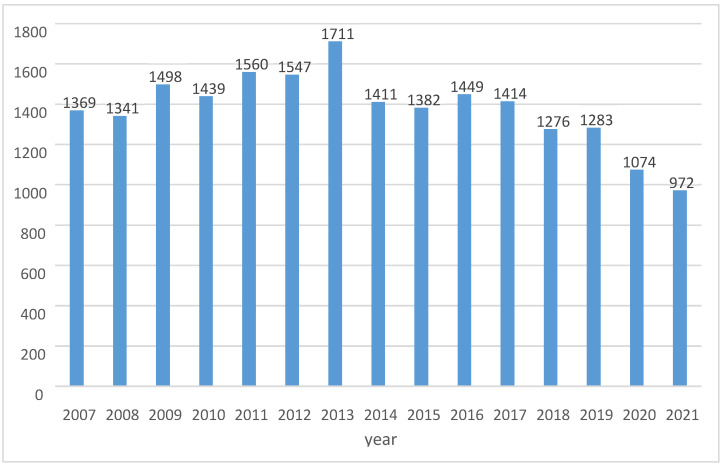
Number of claims filed per year.

**Figure 2 healthcare-10-01929-f002:**
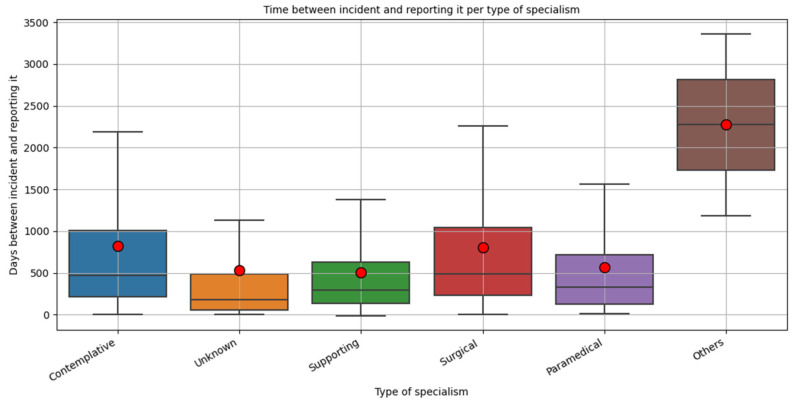
Time between incident and submission of malpractice claims (days). The boxes reflect the first–third quartile. The black lines in the boxes represent the median amount of days between incident and claim submitted. The red dots indicate the mean amount of days.

**Figure 3 healthcare-10-01929-f003:**
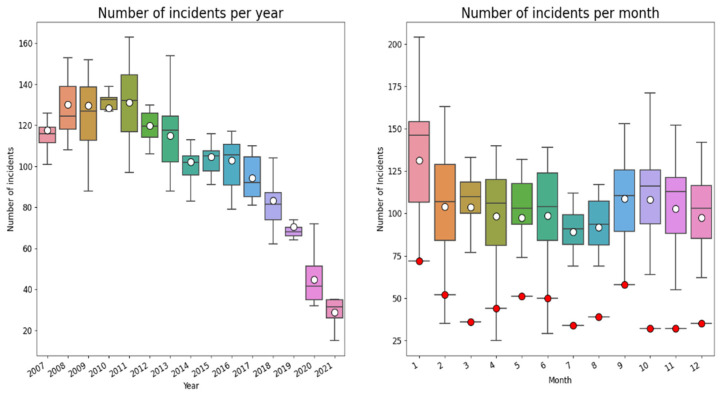
Year and month-wise box plot of reported incidents from 2007–2021. The number of incidents is in both graphs the number of incidents per month. The white dots indicate the mean number of incidents. The red dots are the number of reported incidents in 2020.

**Figure 4 healthcare-10-01929-f004:**
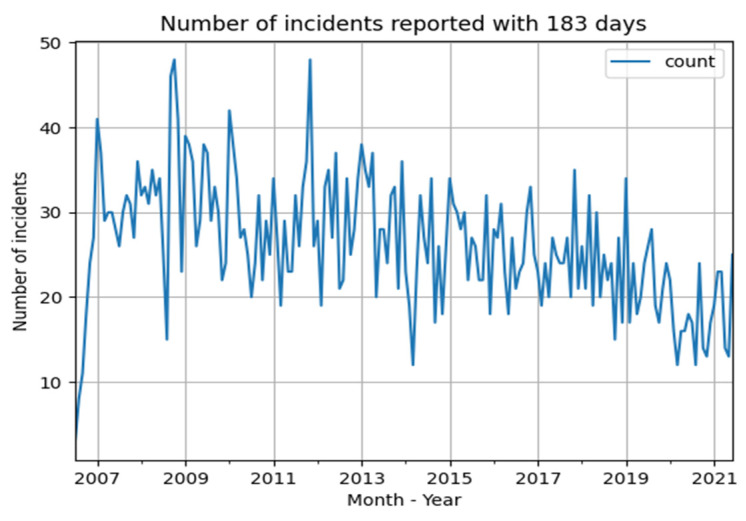
Number of incidents per month, with a reporting time of less than 183 days with date of incident before 1 July 2021.

**Figure 5 healthcare-10-01929-f005:**
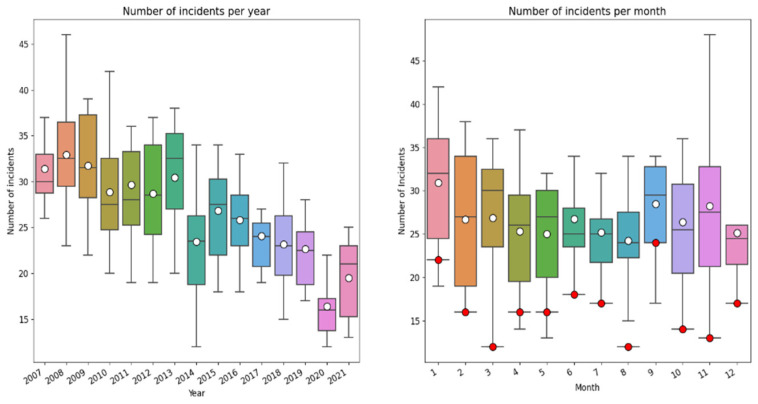
Year and month-wise box plot of reported incidents which were submitted within 183 days after the date of the incident from 2007–2021. The number of incidents is in both graphs the number of incidents per month. The white dots indicate the mean number of incidents. The red dots are the number of reported incidents in 2020.

**Figure 6 healthcare-10-01929-f006:**
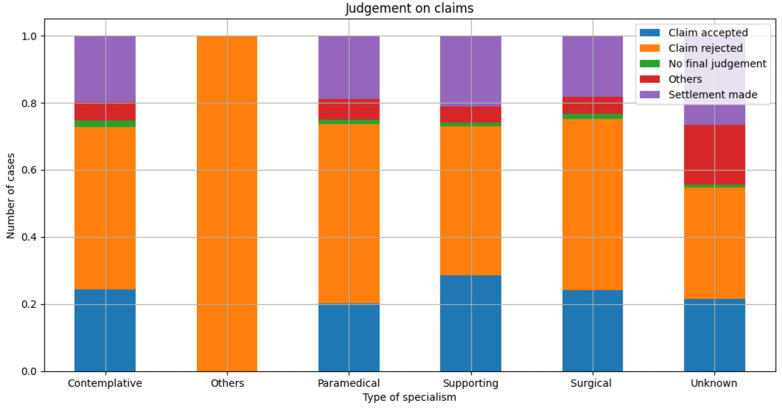
Final judgement per category medical specialty as relative distribution per final judgement.

**Figure 7 healthcare-10-01929-f007:**
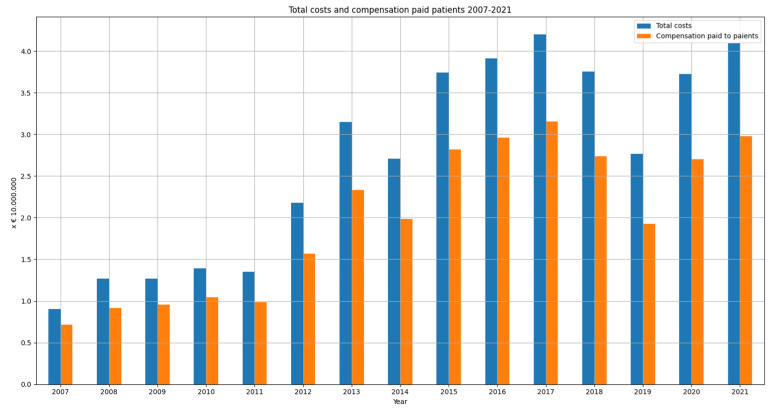
Total costs and compensation paid to the patients (closing year 2007–2021).

**Figure 8 healthcare-10-01929-f008:**
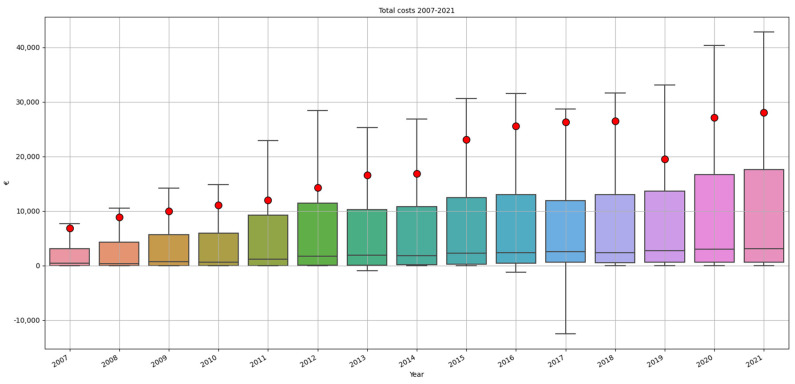
Mean, median, first and third quartile of the total costs per claim per year. The red dots indicate the mean costs, the black lines in the boxes the median costs.

**Table 1 healthcare-10-01929-t001:** Medical specialties per category.

Surgical	Contemplative	Supporting	Paramedical	Unknown	Others
Cardiosurgeon	Fertility doctor	Nurse anesthetist	Occupational therapist	Resident	Others
Surgeon	ER doctor	Anesthesiologist	Physiotherapist	No specialty selected	Social workers
Surgeon, Radiologist	Allergist	Pharmacist	Plastercast nurse	Unknown (yet)	
Gynecologist	Cardiologist	Bacteriologist	Other paramedical professions	Sports doctor	
Oral surgeon	Youth health care physician	Clinical chemist	Podiatrist	Dentist	
ENT doctor	Dermatologist	Laboratory doctor	Psychologist	General hospital doctor	
ENT doctor, neurologist	Gastroenterologist	Lab technician	Midwife	Hospital staff	
Neurosurgeon	Geriatrician	Nuclear doctor			
Ophthalmologist	General practitioner	Nurse practitioner			
Orthopedic surgeon	Intensivist	Pathologist			
Plastic surgeon	Internist	Physician assistant			
Trauma surgeon	Pediatrician	Radiological technician			
Urologist	Pulmonologist	Radiologist			
	Microbiologist	Radiologist, Surgeon			
	Nephrologist	Radiotherapist			
	Neurologist	Nurse			
	Elderly care physician				
	Psychiatrist				
	Rheumatologist				
	Rehabilitation doctor				

**Table 2 healthcare-10-01929-t002:** Top 10 medical specialties with the highest costs (total claim costs in 15 years) related to their share in financial turnover of all medical specialties (the share per medical specialty based on 2012).

Medical Specialties	Total Costs	%	Paid to Patient	%	Financial Turnover	%
Surgeon	€ 99,990,000	24.74%	€ 73,350,000	24.63%	€ 2,026,717,000	15.26%
Orthopedic surgeon	€ 59,290,000	14.67%	€ 41,320,000	13.88%	€ 1,252,586,000	9.43%
Gynecologist/obstetrics	€ 44,440,000	10.99%	€ 34,290,000	11.52%	€ 975,436,000	7.34%
Neurologist	€ 18,820,000	4.66%	€ 14,100,000	4.74%	€ 705,718,000	5.31%
Anesthesiologist	€ 18,190,000	4.50%	€ 14,540,000	4.88%	€ 136,148,000	1.03%
Internist	€ 15,460,000	3.82%	€ 11,790,000	3.96%	€ 1,327,649,000	10.00%
Urologist	€ 13,000,000	3.22%	€ 9,500,000	3.19%	€ 528,728,000	398%
Cardiologist	€ 12,210,000	3.02%	€ 9,040,000	3.04%	€ 1,504,369,000	11.33%
Radiologist	€ 11,030,000	2.73%	€ 8,120,000	2.73%	€ 77,691,000	0.58%
ER doctor	€ 10,310,000	2.55%	€ 7,490,000	2.52%		

## Data Availability

Restrictions apply to the availability of these data. Data were obtained from Centramed and MediRisk and are available on request from the corresponding author, with the permission of Centramed and MediRisk.

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
