# Peer review of "Trends and Developments in Medical Liability Claims in The Netherlands†"

_healthcare, 2022, doi:10.3390/healthcare10101929_

Round 1

Reviewer 1 Report

The report is interesting; however, the authors should make some changes and add some specific clarifications.

1) As rightly pointed out, the authors declare that they are not aware of how many liability claims have been received in court and how much the final costs are, if negligence is proved. This is a big limitation; having this kind of feedback is a great tool for professionals. There is no way to get data, even raw, of how many times Centramed or MediRisk have been sued in court?

2) Another limitation is the lack of verification of a possible simultaneous involvement of several medical specialties, for the same case. Are there, for example, cases that involved both radiology and surgery at the same time? This would be a fundamental report that would open equally important scenarios in the field of team professional liability. If the data cannot be extrapolated, please argue this as a limitation of the study.

3) Is it possible to know how many deaths there have been per year? The cost of a "fatal event" liability is much higher than for other types of negligence. Authors are required to verify and report this data, including with a graph.

4) Is there the possibility of a compensation mechanism by the insurance companies against the specialist who has committed gross negligence? Is there any data?

These questions are asked because the study, although well thought out, lacks a useful feedback tool for good medical practice. Authors should discuss this in the Conclusions section.

Minor tips

1) I would suggest inserting "In Netherlands" in the title

2) The authors support that "In total, 20,726 claims were filed and 21,826 claims were closed". Are those closed in greater number than those filed? Please check.

Reviewer 2 Report

The authors analyzed 15 years of medical malpractice claims in the Netherlands. They claim that all claims were filed and managed by two insurance companies between 2007-2021. Given the previous analysis, the authors highlighted that, since the COVID-19 pandemic, a decrease had been observed in the number of complaints presented and in the number of reported incidents. The results give information on trends and developments in the number and costs of liability claims in hospitals in the Netherlands.

The article discusses an important topic related to medical negligence. This recurring theme is related to a theme that has worried several stakeholders in healthcare management. I have some suggestions:

1 - Why was the Netherlands selected and not another country for the study to be carried out? Justify the choice of this market.

2 - Why were 15 years selected? Justify the choice of this period.

3 - Methodology: To evaluate medical liability claims in the Netherlands, the authors used the databases from Centramed and MediRisk. Detail more about the selected databases.

  - "Group A includes all hospital-related claims filed by Centramed and MediRisk between January 1, 2007, and December 31, 2021. Group B consists of all hospital-related claims closed by Centramed and MediRisk between January 1, 2007, and December 31, 2021 ". Explain better the difference between the two groups.

4 - The authors state that this is the first study to describe the effect of COVID-19 on malpractice claims. In this case, this statement must be justified. For example, only the two years (2020 and 2021) consider the pandemic period. And the other 13 years? From my understanding, it is a fine line to make this statement.

5 - Why does negligence happen in hospitals in the Netherlands? What is the impact of this analysis on other countries? How could other regions benefit from this study?

6 - Highlight this study's theoretical, managerial, and political implications.

7 - Insert the limitations of the research in the conclusions.

Round 2

Reviewer 2 Report

I'm satisfied with the responses. The paper has been improved in its current form.